# Determination of Johnson–Cook Constitutive Parameters for Cutting Simulations

**Michael Storchak \*** , **Philipp Rupp, Hans-Christian Möhring and Thomas Stehle**

Institute for Machine Tools, University of Stuttgart, Holzgartenstraße 17, 70174 Stuttgart, Germany;
philipprupp91@gmail.com (P.R.); hans-christian.moehring@ifw.uni-stuttgart.de (H.-C.M.);
thomas.stehle@ifw.uni-stuttgart.de (T.S.)

**\*** Correspondence: michael.storchak@ifw.uni-stuttgart.de; Tel.: +49-711-685-83831

**Abstract:** The Johnson–Cook constitutive equation is very widely used for simulating cutting processes. Different methods are applied for establishing parameters of the constitutive equation. Based on the methods analysed in this study, two algorithms were worked out to determine the constitutive parameters for the prevailing conditions during cutting processes. In the first algorithm, all constitutive parameters were established simultaneously with standardized test methods. In the second algorithm, the constitutive parameters were established separately in accordance with the cutting conditions prevailing in machining processes. The developed methodology was verified with AISI 1045 heat-treatable steel and Ti10V2Fe3Al (Ti-1023) titanium alloy. The two materials were examined in standardized tensile and compression tests with varying strain rates and temperatures. In addition, the kinetic characteristics of the orthogonal cutting process were established. Based on the results obtained by experiment and the algorithms developed, the constitutive parameters for the cutting conditions were calculated. The parameters were used to determine the material model for simulating the orthogonal cutting process. The algorithms developed were verified by comparing the simulated and experimentally determined kinetic cutting characteristics, which confirmed their good quality.

**Keywords:** computational constitutive modelling and experimental validation; Johnson–Cook constitutive equation; tensile/compression and cutting test; cutting simulation

## 1. Introduction

Numerical calculation methods are gaining more and more importance for the modelling and the simulation of cutting processes [1,2]. Simulations of machining processes are used successfully for the determination of kinetic and other cutting characteristics, such as cutting temperature, chip morphology and chip form, as well as the quality characteristics of boundary layers, such as residual stress [3–5]. Nonetheless the simulation results still differ considerably from the experimental results with regard to the main characteristics of machining processes. As several researchers found out, the main reason for this is that the components of the numerical cutting models can represent real physical-mechanical phenomena in the shear zones only very partially. This is particularly true for those material models or constitutive equations of the work material [6] that comprise the dependences between the deformation of the machined material, the speed of this deformation or strain rate as well as deformation temperature and stresses. The commercial program environments now used for the numerical simulations of different machining processes contain numerous material models [7]. The material models are either stored in the material database of these programs or have to be defined by the user. The latter has to be done very often, especially when the examined material differs from well-known or standardized mechanical values e.g., due to differences in batches or chemical

compositions. In this case, the user is required to define the parameters of the respective material models or constitutive equations, which costs a lot of time and money in principle.

This paper presents the results obtained when establishing the parameters of a constitutive Johnson–Cook equation often used for the simulation of different machining processes [8]. A procedure is suggested here for establishing the parameters of the material model, owing to the experimental and theoretical analyses.

## 2. Establishing the Parameters of the Constitutive Equation

Semi-empirical models such as the Johnson–Cook constitutive equation have become most popular for modelling different machining processes, due to their simplicity and capability to adequately describe flow curves in a wide variation range of the basic parameters [8]. The Johnson–Cook equation is:

$$\sigma_S = (A + B \cdot \varepsilon^n) \cdot \left[1 + C \cdot \ln\left(\frac{\dot{\varepsilon}}{\dot{\varepsilon}_0}\right)\right] \cdot \left[1 - \left(\frac{T - T_0}{T_m - T_0}\right)^m\right], \tag{1}$$

where $\sigma_s$ is the yield point, A is the initial yield stress, B is the stress coefficient of strain hardening, $n$ is the power coefficient of strain hardening, C is the strain rate coefficient, m is the power coefficient of thermal softening, $\varepsilon$ is the strain, $\dot{\varepsilon}$ is the strain rate, $\dot{\varepsilon}_0$ is the reference value of strain rate, $T$ is the actual temperature, $T_0$ is the reference or room temperature and $T_m$ is the melting temperature of the material. The constitutive equation contains only five constants which have to be established by experiment: *A*, *B*, *n*, *C* and *m*. By comparison, other constitutive equations have considerably more constants that must be established [7,9,10].

To determine the constants of the Johnson–Cook constitutive equation to be used in the modelling of cutting processes, different methods are used. They can mainly be divided into two groups: the ones based on experimental determination and those based on inverse identification.

The first method determines the constants, by means of standardized tensile/compression tests or by means of other experimental tests, such as the split Hopkinson pressure bar test [11–13]. The flow curves obtained from the experiments are approximated with regard to the constants of the Johnson–Cook constitutive equation by using well-known methods such as [14].

The methods of the second group were developed in the last two decades and are used especially in the simulation of cutting processes [15–21]. According to the method of determination, the constants of the constitutive equation are corrected by fitting the flow curve. This correction is aimed at achieving a minimum deviation between the simulated and the experimentally obtained characteristics of the same cutting process. The initial values of the constitutive parameters are defined based on published data or by means of standardized mechanical tests carried out specifically for this purpose.

Such a procedure for establishing the constants is used both for the models based on the Johnson–Cook constitutive equation. The new models which have been developed in recent years take account of various physical phenomena in the work material that occur in the boundary layers of the workpiece due to the respective cutting processes. Nemat-Nasser et al. developed a material model taking account of the micromechanical processes during machining, particularly with regard to titanium alloys [22]. This model describes the effects of strain hardening, thermal softening and the so-called dynamic strain ageing. Simultaneously, tests were carried out adapting the Johnson–Cook constitutive equation to the physical processes during cutting processes by means of adding additional terms into the constitutive equation. Furthermore, the Johnson–Cook model was modified to allow for the hardening of the material at great degrees of deformation and supplemented with an additional term describing the temperature-dependent flow softening [23,24]. Based on this development, Sima and Özel worked out a modified material model in which a flow softening phenomenon was linked to strain hardening and thermal softening [25]. Umbrello et al. developed a hardness-based flow stress model to take the hardness of the work material into consideration [26]. The effect of hardness on flow stress was taken into account here with the additional constants J and K supplementing the hardening term. Denguir et al. included further phenomena such as the microstructural transformation effect and

the state of stress effect by supplementing the Johnson–Cook model with two additional terms [27]. Further developments of the models, for example, take account of high strain rates [28], the thermal softening of the work material [29] and other things.

For correcting the constants of the constitutive equations, different algorithms are used, such as a combined algorithm between cutting tests and Oxley's machining theory [16,17], the Levenberg–Marquardt algorithm [18], evolutionary computational [20] and genetic algorithms [30], the response surface methodology [31], etc. Apart from that, a direct simulation-based determination of the material model parameters is very often used by means of a design of experiments (DOE) analysis implemented in some commercial finite element method (FEM) program environments (see e.g., [17,19,21]).

Regarding the identification of the Johnson–Cook constitutive equation by using the inverse method, the five parameters (*A*, *B*, *n*, *C* and *m*) are established either simultaneously [16–18,20] or separately, i.e., first the parameters *A*, *B* and *n* for the respective strain rates and temperatures and then the parameters *C* and *m* [19,32]. Representing the effect of temperature, the parameter *m* is established either simultaneously with the parameter *C* due to the DOE study by comparing simulated and experimentally determined cutting tests [19] or separately by including Oxley's machining theory [32,33].

The paper presents the procedures developed for establishing the constants of the Johnson–Cook constitutive equation by means of conducted standardized compression tests as well as cutting tests.

## 3. Methodology for Identifying the Parameters of Constitutive Equations

The material constants of the Johnson–Cook constitutive equation were established here using two algorithms. In the first algorithm the constants were established simultaneously by adjusting the flow curves obtained by standardized tensile/compression tests. In the second algorithm the parameters *A*, *B*, *n* were determined separately with the flow curves obtained, before the parameters *m* and *C* were identified.

According to the first algorithms, all five constants of the Johnson–Cook constitutive equation were established simultaneously from experimental data of standardized or common tensile/compression tests (see e.g., [11]) by approximating or adjusting the flow curve obtained in the tests. This task is mathematically no single tasking and depends on the assumed initial values of constitutive parameters. Thus various sets of the constants to be established were obtained for their different initial values. This suggested that the adjustment function of the flow curve is a function with several extreme values. In this case the task of establishing a single set of the constants, which corresponds to the global or greatest extreme value of the adjustment function, could be postulated as:

$$\underset{\mathbb{R}^k}{\forall P} \, \underset{\mathbb{R}^k}{\exists S} : G \Rightarrow glextr \, f(X), \, P = \overset{k}{\underset{i=1}{\cup}} S_i, \tag{2}$$

where *P* is the possible amount of parameter sets which consist of the constants *A*, *B*, *n*, *C* and *m* and are defined in a k-dimensional space $\mathbb{R}^k$; *S* is the amount of parameter sets corresponding to the extreme values of the adjustment function f(X); *G* is the parameter set corresponding to the global extreme value of the adjustment function. The second algorithm is based on the approach that has been developed in the last years. In this approach, the individual terms of the Johnson–Cook constitutive equation and the corresponding physical processes are taken into account separately [25,32]. Regarding the classic Johnson–Cook equation, this examination could be expressed as:

$$\underset{\mathbb{R}^k}{\forall P} \, \underset{\mathbb{R}^k}{\exists S} : G \Rightarrow \sigma_S = K_\varepsilon \cdot K_{\dot{\varepsilon}} \cdot K_T, \tag{3a}$$

$$K_\varepsilon = A + B \cdot \varepsilon^n, \, K_{\dot{\varepsilon}} = 1 + C \cdot \ln\!\left(\frac{\dot{\varepsilon}}{\dot{\varepsilon}_0}\right), \, K_T = 1 - \left(\frac{T - T_0}{T_m - T_0}\right)^m, \tag{3b}$$

where $K_\varepsilon$ is the term for strain hardening, $K_{\dot\varepsilon}$ is the term for strain rate sensitivity and $K_T$ is the term for thermal softening. Such a procedure can also be used for establishing the constants of newly developed material models based on the Johnson–Cook constitutive equation [19,21–23,25,26].

In the first step of the second algorithm, the constants $A$, $B$ and $n$, characterizing the term for strain hardening in the Johnson–Cook constitutive equation, were determined by adjusting the flow curves obtained in standardized tensile/compression tests. Thus, the adjusted coefficient of hardening $K_\varepsilon$ was established.

In the second step of this algorithm, the thermal softening power coefficient m was established. The constants $A$, $B$ and $n$ of the strain hardening term, which were established by adjusting the flow curves obtained in tensile/compression tests at varying temperatures, were put in for that purpose. Then the coefficients of correction were established for every temperature used in these tests by means of the following equation [32]:

$$K_T(\varepsilon) = \frac{\sigma_S^T(\varepsilon)}{\sigma_S^{T_0}(\varepsilon)}. \tag{4}$$

An average coefficient of correction $K_T$ was calculated for every tensile/compression test at the respective temperature. Then the coefficient m was established by approximating (adjusting) the calculated range of average correction coefficients with the following equation [32]:

$$K_T(T) = 1 - \left(\frac{T - T_0}{T_m - T_0}\right)^m. \tag{5}$$

The constant $C$ of the strain rate sensitivity term $K_\varepsilon$ was determined by means of Oxley's machining theory [33] and the extension of Oxley's machining theory [16,34–37], using kinetic data obtained by orthogonal cutting experiments:

$$C = \frac{1 - \frac{\sigma_S}{K_\varepsilon \cdot K_T}}{\ln\left(\frac{\dot\varepsilon}{\dot\varepsilon_0}\right)}. \tag{6}$$

Hence, it was possible to establish the constant $C$ at the same strain rate as well as under the same conditions of the work material's hardening and softening like in a real cutting process. The equivalent shear strain $\varepsilon_{AB}$ and the equivalent strain rate $\dot\varepsilon_{AB}$ in the shear zone AB or primary shear zone were determined according to Oxley [33]:

$$\varepsilon_{AB} = \frac{\cos\gamma}{2 \cdot \sqrt{3} \cdot \cos(\phi - \gamma) \cdot \sin\phi}, \tag{7}$$

$$\dot\varepsilon_{AB} = C_0 \cdot \frac{V_C}{\sqrt{3} \cdot l_{AB}}, \quad V_C = V \cdot \frac{\cos\gamma}{\cos(\phi - \gamma)}, \tag{8}$$

where $\Phi$ is the shear angle, $\gamma$ is the tool orthogonal rake angle of the tool wedge, $C_0$ is the thickness ratio of the primary shear zone, $l_{AB}$ is the length of the shear plane, $V_C$ is the chip speed and $V$ is the cutting speed. When the chip compression ratio $K_a$ or the chip thickness $a_c$ is determined by experiment, then the strain $\varepsilon_{AB}$ can be established with the following equation [38,39]:

$$\varepsilon_{AB} = \frac{K_a^2 - 2 \cdot K_a \cdot \sin\gamma + 1}{\sqrt{3} \cdot K_a \cdot \cos\gamma}. \tag{9}$$

The yield point $\sigma_s$ was equated with the shear stress $\tau_{AB}$ in the primary shear zone along the conditional shear plane $AB$ [16,35,36]:

$$\sigma_S = \tau_{AB} = \frac{F_{AB} \cdot \sin\phi}{a \cdot w}, \tag{10}$$

where $F_{AB}$ is the shear force along the shear plane *AB*, *a* and *w* are the undeformed chip thickness (cutting depth) and the cutting width, respectively. The shear force $F_{AB}$ was calculated based on the measured cutting force $F_X$ and thrust force $F_Z$ [16,40,41]:

$$F_{AB} = F_X \cdot \cos \phi - F_Z \cdot \sin \phi \tag{11}$$

The shear angle $\Phi$ was determined either by experiment due to the measured chip thicknesses $a_C$ (see Equation (12)) [16] and the chip compression ratio $K_a$ [41] (see Equation (13)) or analytically [40]:

$$\phi = \arctan\left( \frac{\frac{a}{a_C} \cdot \cos \gamma}{1 - \frac{a}{a_C} \cdot \sin \gamma} \right), \tag{12}$$

$$\frac{a_C}{a} = K_a. \tag{13}$$

The temperature $T_{AB}$ which arises due to the plastic deformation in the primary shear zone and is used for establishing the coefficients of correction $K_T$ can be determined with the following equation [16,33,36,42]:

$$T_{AB} = T_0 + \eta \cdot \frac{1 - \beta}{\rho \cdot S \cdot a \cdot w} \cdot \frac{F_{AB} \cdot \cos \gamma}{\cos(\phi - \gamma)}, \tag{14}$$

$$\begin{cases} \beta = 0.5 - 0.35 \cdot \log_{10}(R_T \cdot \tan \phi) \ for \ 0.04 \leq R_T \cdot \tan \phi \leq 10.0 \\ \\ \beta = 0.3 - 0.15 \cdot \log_{10}(R_T \cdot \tan \phi) \ for \ R_T \cdot \tan \phi > 10.0 \end{cases}, \tag{15}$$

$$R_T = \frac{\rho \cdot S \cdot V \cdot a}{\lambda}, \tag{16}$$

where $T_0$ is the room temperature, $\eta$ is the parameter to scale the average temperature rise at *AB* $(0 < \eta \leq 1)$, $\beta$ is the proportion of heat conducted into the workpiece, $\rho$ is the density, *S* is the specific heat, $R_T$ is the non-dimensional thermal number, $\lambda$ is the thermal conductivity of the work material. This task was solved with the algorithms developed for that purpose, as described in Section 6.

## 4. Test Set-Up

The material properties were established with standard tensile and compression tests of the material. The machining process was examined in orthogonal cutting experiments. AISI 1045 heat-treatable steel and Ti10V2Fe3Al (Ti-1023) titanium alloy were used as test materials. The mechanical and thermal properties of the two materials are listed in Table 1. The materials were stress-relieved before the tests.

**Table 1.** Mechanical and thermal properties of test materials [4,43].

| Material | Strength (MPa) | | Elastic Modulus (GPa) | Elongation (%) | Hard-ness (HB) | Poisson's Ratio | Specific Heat (J/kg·K) | Thermal Expansion (μm/m·°C) | Thermal Conductivity (W/m·K) |
|---|---|---|---|---|---|---|---|---|---|
| | Tensile | Yield | | | | | | | |
| AISI 1045 | 690 | 620 | 206 | 12 | 180 | 0.29 | 486 | 14 | 49.8 |
| Ti-1023 | 1282 | 1220 | 110 | 4–10 | 369 | 0.33 | 527 | 9.7 | 7.8 |

*4.1. Tensile/Compression Tests*

### 4.1.1. Tensile Tests

The tensile tests were carried out with the universal testing machine Roell + Korthaus 100 (ZwickRoell GmbH & Co. KG, Ulm, Germany), guaranteeing a maximum load of 100 kN, in accordance with standard DIN EN ISO 6892-1 and 6892-2. The experiments were carried out at room temperature and varying strain rates of 0.013 s$^{-1}$, 0.067 s$^{-1}$ and 0.12 s$^{-1}$.

### 4.1.2. Compression Tests

The compression tests for establishing the material properties were carried out with the Gleeble System 3800 (Dynamic Systems Inc., NY, USA) in accordance with the standard DIN 50106. For that purpose, cylindrical test specimens with an 8 mm diameter and a 12 mm height were produced and compressed at strain rates of 0.05 s$^{-1}$, 1 s$^{-1}$, 10 s$^{-1}$ and varying temperatures. In the tests, the examined specimens had varying temperatures: room temperature (20 °C) as well as 200 °C, 400 °C, 600 °C and 800 °C. Each measurement was repeated several times to guarantee a certain reliability of the data measured and to compensate for an existing scattering in the composition as well as the material properties of the test materials. Then flow curves of the respective material were chosen in order to establish the parameters of the constitutive equation.

### 4.2. Cutting Tests

The orthogonal cutting processes of AISI 1045 heat-treatable steel and Ti-1023 titanium alloy were analysed by experiment using a special test stand [4,41]. The tool was stiffly clamped onto the test stand. The indexable insert was clamped in the tool rotating head at a particular tool orthogonal rake angle by means of a clamping jaw [4]. The angle of point was produced by grinding the flank face of the wedge at a corresponding tool orthogonal clearance, using a tool grinding machine. The tool orthogonal clearance of the wedge was 8° in all experimental tests of orthogonal cutting. The wedge edge radius of insert in those experimental tests was 20 μm. The workpiece for orthogonal cutting was out of AISI 1045 and Ti-1023 and had a length of 170 mm, a height of 60 mm and a width of 3 mm, used as width of chip. The workpiece out of the work material was clamped in a three-component dynamometer, type 9263, by Kistler, making it possible to measure two components of the resultant force, namely cutting force $F_X$ and thrust force $F_Z$. Indexable inserts, type SNMG-SM-1105, by Sandvik Coromant were used as inserts.

## 5. Results of Experiments

### 5.1. Tensile/Compression Tests

Figure 1 presents, as an example, those flow curves that were obtained at room temperature and varying strain rates in standardized tensile tests of Ti-1023 titanium alloy. The maximum true stress values of the plastic areas range from about 1100 MPa to 1300 MPa.

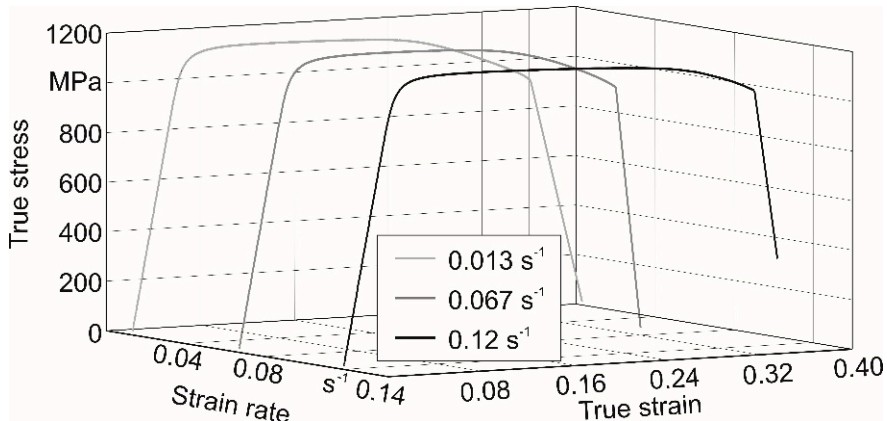

**Figure 1.** Flow curves of Ti-1023 titanium alloy in standardized tensile tests at different strain rates.

The values of elongation are between 6% and 10% of the true strain. When analyzing the individual test specimens in the tensile test, it could be found out that the true elongations of the specimens are in the low to medium range of the true stress for a strain rate of 0.12 s$^{-1}$, in the medium range for a strain rate of 0.067 s$^{-1}$ and in the medium to high range in the case of the remaining test specimens. The

strain rate hardly affected the course of the flow curves within the examined boundaries. This effect would be substantially clearer in the case of considerably higher strain rates. However, this requires the use of very expensive equipment, such as the split Hopkinson pressure bar [6].

In addition to tensile tests, compression tests were also conducted at varying strain rates and temperatures. As an example, Figure 2 presents the flow curves obtained in the standardized compression tests of the AISI 1045 steel specimens at varying temperatures. It shows that there is a great decrease in yield stress with growing temperature as expected. Like in the tensile tests, the strain rate hardly affected the course of the flow curves in the compression tests as well, irrespective of the fact that the strain rate was in a considerably greater range from $0.05 \, \text{s}^{-1}$ to $10 \, \text{s}^{-1}$. In all probability, the strain rate should change much more in proportion to real cutting processes in order to have a noticeable effect.

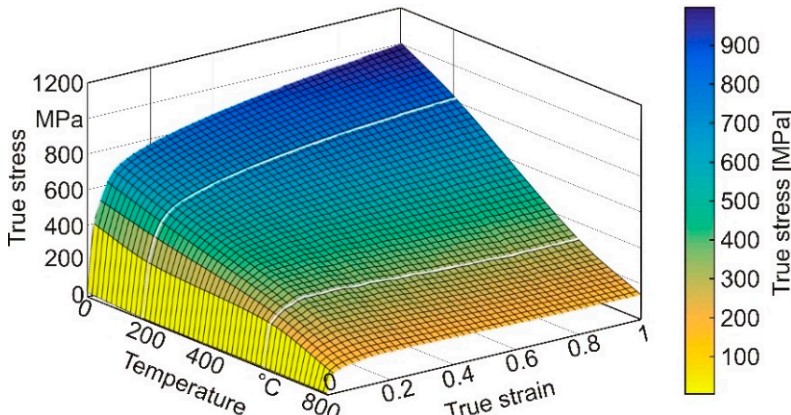

**Figure 2.** Flow curves of AISI 1045 steel in standardized compression tests at different temperatures.

The flow curves obtained in standardized tensile/compression tests were used for establishing the parameter set of the constitutive equation according to the first algorithm (see Section 3), the parameters *A*, *B* and *n* of the strain hardening term as well as the parameter *m* of the thermal softening term.

*5.2. Cutting Tests*

The experimental tests of orthogonal cutting were carried out with three varying parameters: cutting speed, cutting depth (undeformed chip thickness) and tool orthogonal rake angle. In the machining of Ti-1023 titanium alloy, the cutting speed was varied here in four steps: 32 m/min, 48 m/min, 64 m/min and 96 m/min. The cutting depth was changed in three steps: 0.05 mm, 0.1 mm and 0.15 mm. In the machining of AISI 1045 steel, the cutting speed was varied in three steps of 48 m/min, 96 m/min and 144 m/min, whereas the depth of cut or undeformed chip thickness was varied in three steps of 0.1 mm, 0.15 mm and 0.2 mm. The tool orthogonal rake angle of the tool wedge was changed in steps of 10°: −10°, 0° and 10°. Hence, a multifactorial experiment was conducted here with a total of 36 averaged tests in the machining of Ti-1023 titanium alloy and 27 averaged tests in the machining of AISI 1045 steel. For cutting force and chip compression ratio of AISI 1045 was the average measurement uncertainty 8% and 6% corresponding, for cutting force and chip compression ratio of titanium alloy Ti-1023 was the average measurement uncertainty 11% and 10%.

As an example, Figures 3 and 4 present the cutting force $F_X$ at a cutting speed of 96 m/min in the machining of Ti-1023 titanium alloy and AISI 1045 steel, depending on the tool orthogonal rake angle and the depth of cut.

The cutting force $F_X$ increases with growing depth of cut, which can be explained by an increase in the material removal rate. At the same time, the cutting force decreases with growing tool orthogonal rake angle. This can be explained by the decrease in the degree of deformation with growing tool orthogonal rake angle.

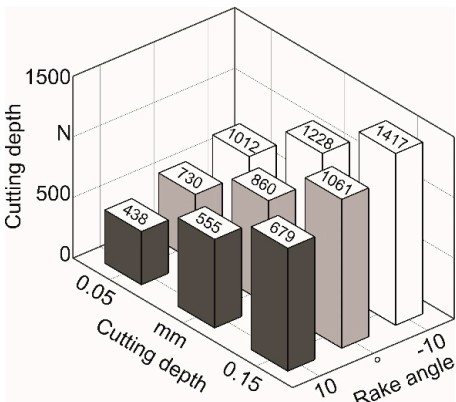

**Figure 3.** Plot of cutting force for different cutting depths and rake angles in the machining of Ti-1023 titanium alloy.

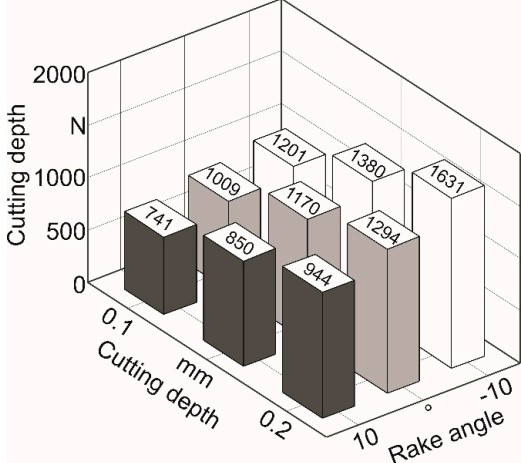

**Figure 4.** Plot of cutting force for different cutting depths and rake angles in the machining of AISI 1045 steel.

As an example, Figure 5 shows the chip compression rate determined in the machining of AISI 1045 steel at an undeformed chip thickness of 0.2 mm, depending on tool orthogonal rake angle and cutting speed. On the whole, it can be summed up that the chip compression ratio decreases with growing cutting speed and increases with decreasing tool orthogonal rake angle.

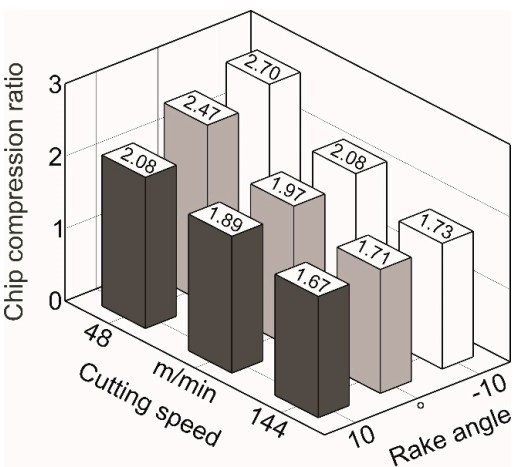

**Figure 5.** Plot of chip compression ratio for different rake angles and cutting speeds in the machining of AISI 1045 steel.

As an example, Figure 6 presents the chip compression ratio determined in the machining of Ti-1023 titanium alloy at an undeformed chip thickness of 0.15 mm, depending on tool orthogonal rake angle and cutting speed. Analogously to the dependence of the chip compression ratio in the machining of AISI 1045 steel, the coefficient here decreases with growing cutting speed and increases with decreasing tool orthogonal rake angle. The absolute value of the chip compression ratio is less than 1 here. This can be observed very often in the machining of titanium alloys, particularly at cutting speeds that are relatively high for these alloys. The reason for that may be the toughness as well as the low thermal conductivity of titanium alloys and, because of that, a higher cutting temperature than in the machining of ferruginous materials [4,25].

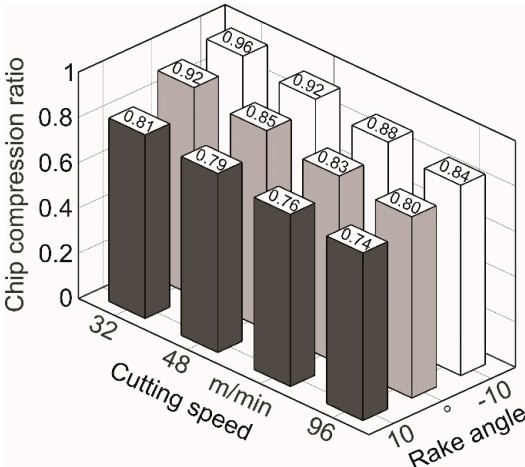

**Figure 6.** Plot of chip compression ratio for different rake angles and cutting speeds in the machining of Ti-1023 titanium alloy.

The experimental data obtained in the cutting tests served for establishing the parameter C of the constitutive equation's strain rate sensitivity term as well as for verifying the simulation model.

## 6. Determination of Constitutive Parameters

All five constitutive parameters were established simultaneously with the first algorithm (see Section 3) by approximating or adjusting the flow curves obtained in the tensile/compression tests. This determination was carried out in two variants: with fixed and free parameter *A*-initial yield stress. In the first variant, the parameter *A* was equated with the yield point Re of the material to be examined, whereas it was adjusted freely together with the other four parameters in the second variant. The coefficient of determination was highest for the free parameter *A* (second variant). This suggested that there was a better adaptation of the adjusted constitutive parameters to the experimental flow curve. In addition, the adjustment of the parameter A to the yield point Re was most realistic.

The analysis of several different approximation algorithms showed a great dependence of the adjusted parameter values on the starting point. This difficulty could be solved with the multistart algorithm [44], which was used successfully for finding the global optimum. Using the multistart algorithms for the two variants (see above) guaranteed that the constitutive parameters could be established with a coefficient of determination greater than 99.5%. Figure 7 depicts the flow chart for the simultaneous determination of constitutive parameters.

The algorithm was, however, restricted by the conditions under which the experimental flow curve, used for the approximation, was obtained. When the conditions of the test method for establishing the flow curve were changed, such as ambient temperature or strain rate, even the material itself or its physical-mechanical condition, the values of the constitutive parameters differed considerably from those determined previously. By the way, this also applied to the parameter values determined with

the inverse method (see e.g., [17–19,21]) due to a fixed value of the cutting characteristics obtained by experiment.

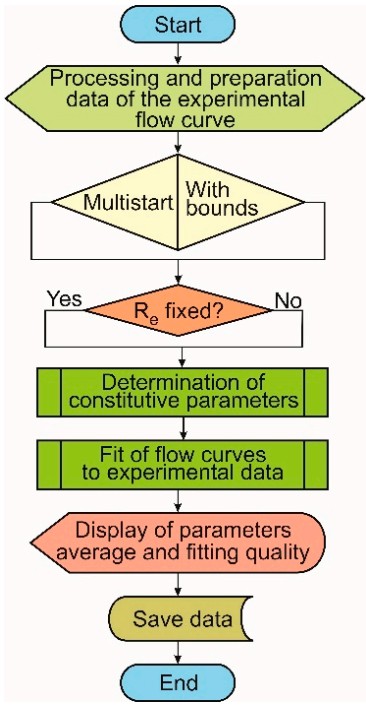

**Figure 7.** Flow chart of algorithm for the simultaneous determination of constitutive parameters.

The determination of the constitutive parameters with the second algorithm (see Section 3) was conducted separately for the different terms of the Johnson–Cook constitutive equation. Figure 8 depicts the flow chart for the separate determination of constitutive parameters.

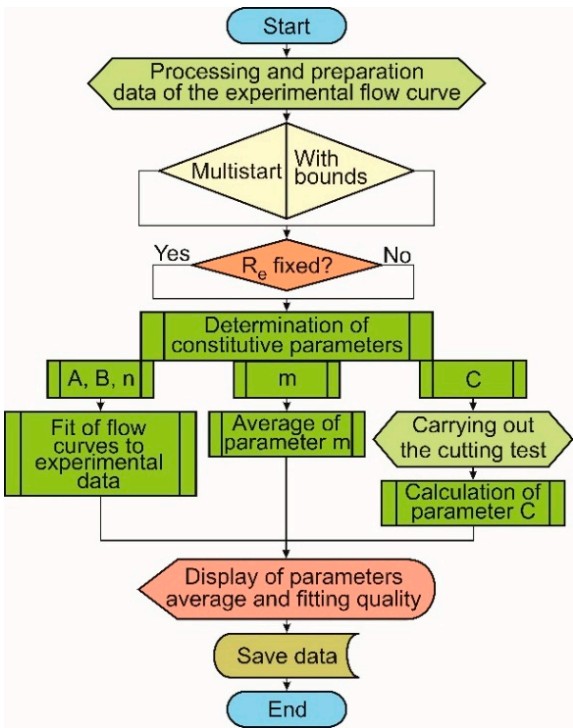

**Figure 8.** Flow chart of algorithm for the separate determination of constitutive parameters.

The constitutive parameters for the two analysed materials (see Section 4) were established based on the algorithm developed (see Section 3 and Figures 7 and 8). These parameters are given in Table 2.

**Table 2.** Constitutive parameters as established with the second algorithm.

| Material | Algorithm | Constitutive Parameters | | | | |
|---|---|---|---|---|---|---|
| | | *A* (MPA) | *B* (MPA) | *n* (-) | *C* (-) | *m* (-) |
| AISI 1045 | First (simultaneously) | 512.3 | 671.7 | 0.2905 | 0.01244 | 1.26 |
| | Second (separately) | 439.125 | 475.948 | 0.2136 | 0.0181201 | 0.848 |
| Ti-1023 | First (simultaneously) | 1012.5 | 875.4 | 0.2613 | 0.0215 | 1.07 |
| | Second (separately) | 976.9 | 502.3 | 0.22 | 0.028 | 0.8 |

Subsequently, the parameters listed in Table 2 were used for creating the material model for the numerical simulation of orthogonal cutting processes.

## 7. Simulation Results

The constitutive parameters calculated using the second (separately) algorithm (see Section 6) were verified with the simulation of an orthogonal cutting process. For that purpose, a 3D cutting model in the FEM software environment DEFORM 2D/3D™ v. 11.0 (SFTC, Columbus, OH, USA) was worked out for simulating the machining of AISI 1045 steel and Ti-1023 titanium alloy. Figure 9 presents the meshed geometrical model.

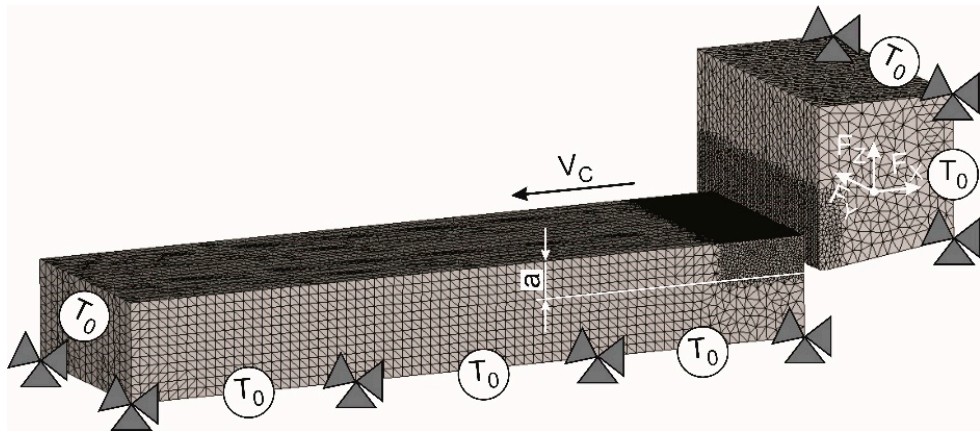

**Figure 9.** Initial geometry and boundary conditions of orthogonal cutting model.

To improve the efficiency and the precision of the cutting model, the workpiece was meshed more finely in the shear zones as well as in the chip and the tool was also meshed more finely in the contact areas with the workpiece and the chip. The mesh in the remaining areas was established more roughly. The boundary conditions were determined by fixing the workpiece and the tool as well as by the input of the thermal conditions at the boundaries of the respective objects. The bottom of the workpiece was rigidly fixed in the *X*-, *Y*- and *Z*-directions. The rigid fixation of the tool at the back of its rake face in *Z*-direction prevents its meshing in this direction. The thermal initial conditions at room temperature (RT) were given at the bottom and the left-hand side of the workpiece as well as at the left-hand side and the top of the tool. The working motion of the tool at a cutting speed VC for guaranteeing the cutting process was given by the absolute motion in the negative *X*-direction.

In order to reproduce the cutting process of the titanium alloy, the Cocroft and Latham model [4] was used as fracture model and the critical breaking stress was set to 240 MPa. Using a DOE (design of experiments) analysis, it was determined that the breaking stress and the load-carrying capacity of the finite elements were 10% after the fracture. The friction during the contact between the tool and the workpiece was modelled with a hybrid friction model. The model represented a combination of the Coulomb and the shear friction models. Regarding the modelling of the cutting process of AISI 1045 steel, the Coulomb friction coefficient was 0.15 and the plastic friction coefficient of the shear friction model was 0.6. These values were 0.2 and 0.8, respectively, when modelling the cutting process of Ti-1023 titanium alloy.

The functioning of the FEM cutting model was verified by simulating the chip formation, the development of the strain rate in the shear zones as well as the temperature flows in the workpiece during the machining of AISI 1045 steel and Ti-1023 titanium alloy. As an example, Table 3 presents the cutting characteristics for different process parameters. The analysis of the simulation results indicated that the FEM cutting model was suitable for the further modelling.

**Table 3.** Exemplary simulation results of the chip morphology, the strain rate in the shear zones and the temperature flows in the workpiece.

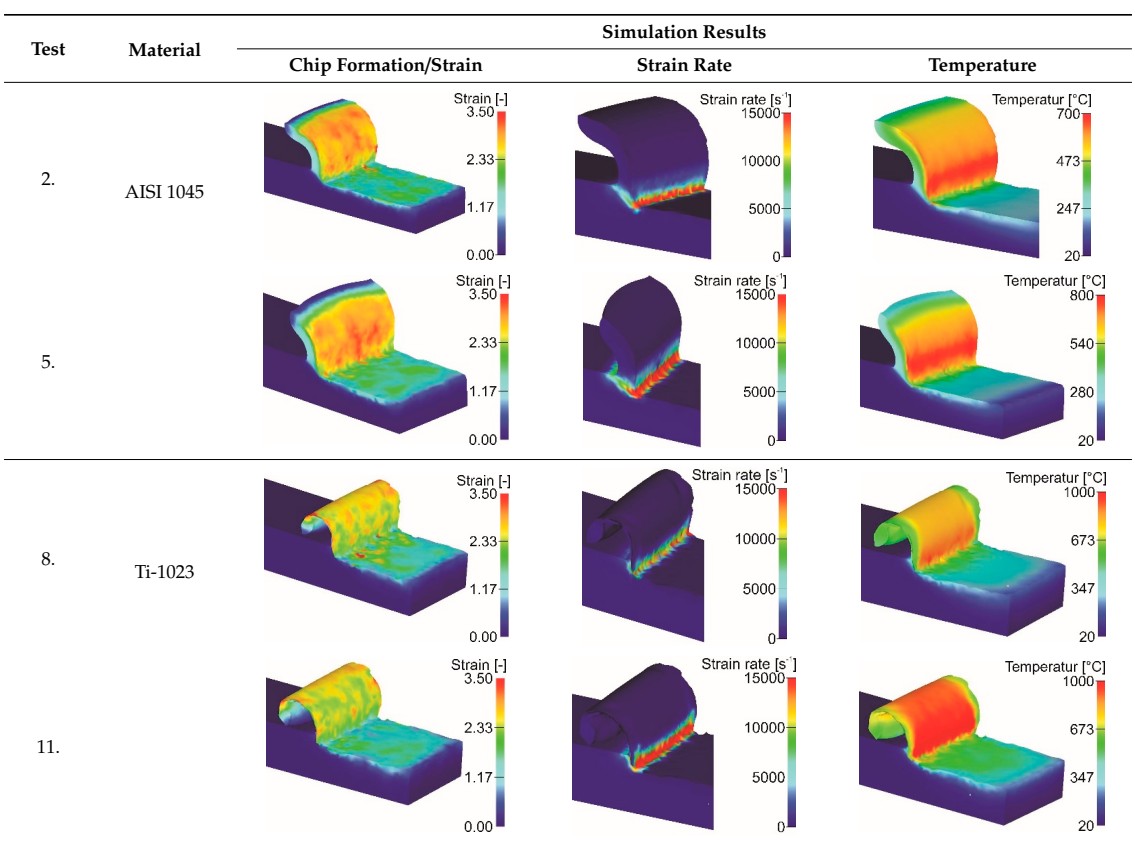

The elaborated FEM cutting model and the quality of the developed algorithm for establishing constitutive parameters were verified by means of comparing the resultant forces determined by experiment as well as the cutting temperatures calculated analytically with the corresponding simulated process characteristics. Table 4 presents the cutting parameters and the tool orthogonal rake angles used in the experimental tests of the machining process for verifying the FEM cutting model.

**Table 4.** Cutting parameters and tool orthogonal rake angles in the experimental and simulative cutting tests.

| Test | Material | Cutting | | Rake Angle |
|------|----------|---------|---------|------------|
| | | Speed (m/min) | Depth (mm) | (°) |
| 1. | | | | −10 |
| 2. | | 96 | | 0 |
| 3. | AISI 1045 | | 0.2 | 10 |
| 4. | | | | −10 |
| 5. | | 144 | | 0 |
| 6. | | | | 10 |
| 7. | | | | −10 |
| 8. | | 48 | | 0 |
| 9. | Ti-1023 | | 0.15 | 10 |
| 10. | | | | −10 |
| 11. | | 96 | | 0 |
| 12. | | | | 10 |

Table 5 shows the results of the comparison between the simulated and the calculated as well as experimental values of the cutting temperature, the resultant forces and the strain in the chip forming area. The comparison (s. the values of deviation) showed a good agreement and confirmed the validity of the elaborated algorithm as well as the FEM cutting model.

**Table 5.** Comparison of predicted and experimental forces in orthogonal cutting.

| Test | Material | Shear Angle | Strain | | Deviation | Cutting Temperature | | Deviation | Cutting Force | | Deviation |
|------|----------|-------------|--------|-------|-----------|---------------------|-------|-----------|---------------|------|-----------|
| | | (°) | (-) | | [%] | (°C) | | (%) | $F_X$ (N) | | (%) |
| | | Calc. | Calc. | Sim. | | Calc. | Sim. | | Sim. | Exp. | |
| 1. | | 23.6 | 1.705 | 1.563 | 8.33 | 345.6 | 305.3 | 11.7 | 1425.7 | 1631.2 | 12.6 |
| 2. | | 26.9 | 1.429 | 1.321 | 7.56 | 279.7 | 240.4 | 14.1 | 1154.6 | 1379.9 | 16.3 |
| 3. | AISI 1045 | 29.9 | 1.213 | 1.093 | 9.89 | 237.5 | 207.8 | 12.5 | 1106.8 | 1201.1 | 7.9 |
| 4. | | 27.3 | 1.557 | 1.387 | 10.92 | 308.3 | 278.6 | 9.6 | 1321.8 | 1567.3 | 15.7 |
| 5. | | 30.3 | 1.325 | 1.185 | 10.57 | 275.4 | 245.8 | 10.7 | 1216.4 | 1459.0 | 16.6 |
| 6. | | 33.3 | 1.127 | 1.023 | 9.23 | 220.1 | 203.4 | 7.6 | 1187.9 | 1314.6 | 9.6 |
| 7. | | 42.0 | 1.380 | 1.236 | 10.43 | 616.1 | 513.6 | 16.6 | 1045.7 | 1274.2 | 17.9 |
| 8. | | 48.7 | 1.164 | 1.056 | 9.28 | 499.0 | 469.5 | 5.9 | 1013 | 1207.5 | 16.1 |
| 9. | Ti-1023 | 57.9 | 1.002 | 0.921 | 8.08 | 639.8 | 463.5 | 27.6 | 968.4 | 1099.3 | 11.9 |
| 10. | | 44.2 | 1.394 | 1.267 | 9.11 | 741.7 | 627.5 | 15.4 | 1238.6 | 1416.9 | 12.6 |
| 11. | | 51.3 | 1.184 | 1.066 | 9.97 | 616.3 | 485.7 | 21.2 | 1046 | 1227.6 | 14.8 |
| 12. | | 60.1 | 1.022 | 0.947 | 7.34 | 613.0 | 472.3 | 23.0 | 927.3 | 1012.4 | 8.4 |

The comparison of the two developed algorithms (see Section 6) was based on the comparison of the simulated and experimental cutting forces. Figure 10 presented this comparison. The Figure 10a,b show the experimental and simulative data when cutting steel AISI 1045 at 96 m/min and 144 m/min, the Figure 10c,d show the results of cutting titanium alloy Ti-1023 at the cutting speeds of 48 m/min and 96 m/min, respectively. The deviation between experimental and simulatively defined cutting forces with the constitutive parameters determined by the first (simultaneous) algorithm is in the range of 17% to 30% for machining the steel AISI 1045 and in the range of 22% to 30% for machining the titanium alloy Ti-1023. This deviation in the constitutive parameters determined used the second (separately) algorithm is in the range of 8% to about 17% for the machining of the AISI 1045 steel and in the range of 8% to 18% in the machining of the titanium alloy Ti-1023 (s. Figure 10 and Table 5).

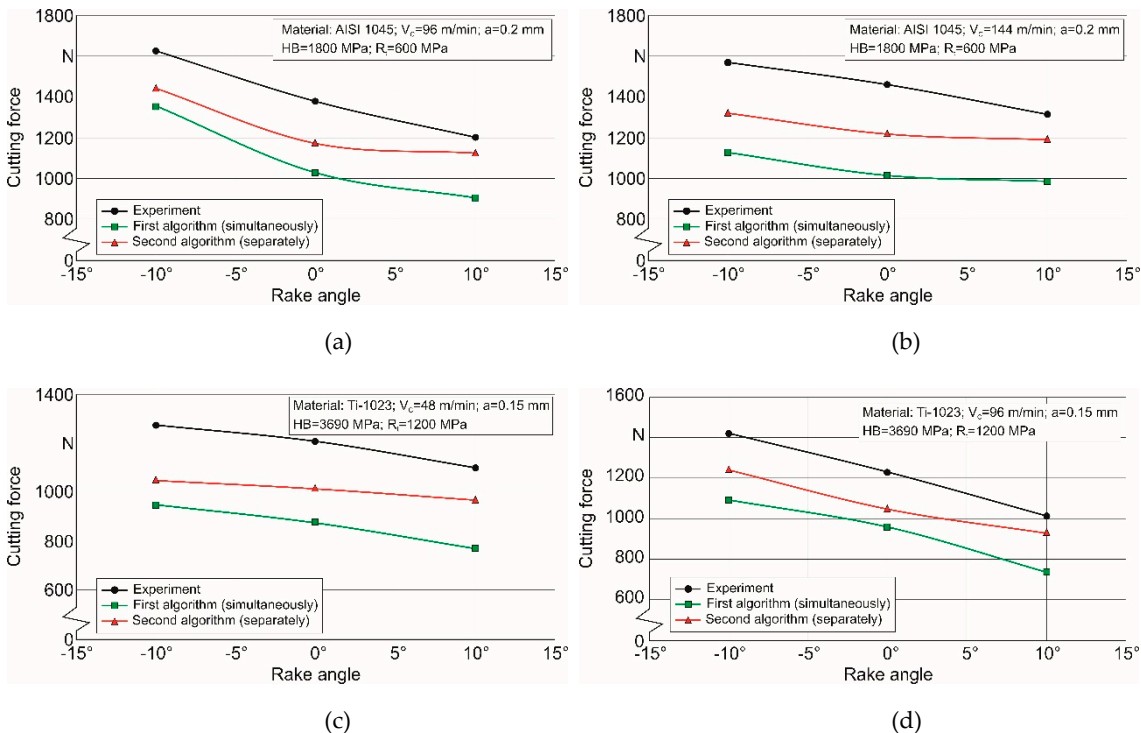

**Figure 10.** Dependence of the cutting force on the rake angle for different cutting speeds ((**a**)—experimental and simulative data when cutting steel AISI 1045 at 96 m/min, (**b**)—the same by cutting speed 144 m/min, (**c**)—experimental and simulative data when cutting titanium alloy Ti-1023 at the cutting speeds of 48 m/min, (**d**)—the same by cutting speed 96 m/min).

The much larger deviation in the experimentally and simulatively determined cutting forces using the first (simultaneous) algorithm is in all likelihood that the first algorithm only ensured the numerical conditions of the fitting. The algorithm does not account for the real physical processes involved in machining such as softening of the material being processed due to thermal action, and additional material hardening due to the increase in strain rate with increasing cutting speed. The physical processes are taken into account in determining the constitutive parameters using the second algorithm, since the different parameters such as m and C are defined under the same conditions as in a real cutting process. As a result, the second algorithm provided significantly better agreement between the experimental and simulatively determined cutting forces.

## 8. Conclusions

In the study presented here, a methodology was worked out to establish constitutive parameters for the conditions prevailing during cutting processes. It was realized with two algorithms. In the first algorithm, all Johnson–Cook constitutive parameters were established simultaneously by means of standardized test methods. By implementing the multistart method in the algorithm, it was possible to guarantee that the constitutive parameters or the global extreme values could be determined unambiguously. The algorithm could be used successfully for establishing the constitutive parameters of occasional process parameter sets.

In the second algorithm, the constitutive parameters were established separately. The determination took account of the physical processes prevailing in a machining process. In this way, the parameter *m* was calculated by adjusting the flow curves obtained in compression tests at varying temperatures. The parameter *C* was established in cutting tests and, correspondingly, at real strain rates. Hence, the constitutive parameters were determined under those condition that these parameters should model according to the Johnson–Cook constitutive equation.

The constitutive parameters determined with the second algorithm were used for the numerical modelling of an orthogonal cutting process. There was a good agreement between the calculated results obtained by experiment and the simulation results, which were gained at different cutting parameters and geometries of the wedge.

**Author Contributions:** Conceptualization, M.S.; Methodology, M.S.; Software, M.S. and P.R.; Validation, M.S.; Formal analysis, M.S. and P.R.; Investigation, M.S. and P.R.; Resources, M.S. and P.R.; Data curation, M.S. and P.R.; Writing—original draft preparation, M.S.; Writing—review and editing, M.S., H.-C.M. and T.S.; Visualization, M.S.; Supervision, H.-C.M. and T.S.; Project administration, H.-C.M. and T.S.; Funding acquisition, H.-C.M. and T.S.

**Funding:** This study was funded by the German Research Foundation (DFG) in the project HE-1656/153-1 "Development of a Concept for Determining the Mechanical Properties of the Cutting Material in Machining".

**Acknowledgments:** The authors would like to thank the German Research Foundation (DFG) for their support, which is highly appreciated.

**Conflicts of Interest:** The authors declare no conflict of interest.

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
