# Peer review of "Determination of Johnson–Cook Constitutive Parameters for Cutting Simulations"

_metals, doi:10.3390/met9040473_

Reviewer 1 Report

The manuscript requires important corrections from the point of view of the logic flow of information and from that of the formulation of the statements. Some difficulties are highlighted in the uploaded annotated pdf file.

Author Response

Answers to the comments of Reviewer #1:

The authors are very grateful to Reviewer # 1 for meticulously reviewing and interpreting the content of the paper.

Reviewer #1: The manuscript requires important corrections from the point of view of the logic flow of information and from that of the formulation of the statements. Some difficulties are highlighted in the uploaded annotated pdf file.

1.       “The manuscript requires important corrections from the point of view of the logic flow of information and from that of the formulation of the statements.”

All suggestions of the reviewer are considered and corrected in the text of the manuscript.

2.       “The flow curve of the material or the first approximation of the constants to be established are taken from literature or the standardized mechanical tests conducted. What does it mean?”

For carrying out the inverse identification, the initial values of the constitutive parameters are to be defined. They are defined based either on published data or from standardized mechanical tests carried out specifically for this purpose.

The sentence: „The flow curve of the material or the first approximation of the constants to be established are taken from literature or the standardized mechanical tests conducted “ is reformulated in the manuscript: “The initial values of the constitutive parameters are defined based on published data or by means of standardized mechanical tests carried out specifically for this purpose.”

3.       “… arise in the boundary layers of the component due to the respective cutting processes. What does it mean?”

The high mechanical and thermal loads in the contact between the tool flank face and workpiece during machining lead to significant changes in the boundary layers of the workpiece. These changes are expressed by z. B. hardening and formation of residual stresses in boundary layers of the workpiece. The sentence: “… arise in the boundary layers of the component due to the respective cutting processes” is reformulated in the manuscript: “… occur in the boundary layers of the workpiece due to the respective cutting processes.”

4.       “… was not mathematically distinctive, … . What does it mean?”

The fitting of the flow curve with 5 constitutive parameters at the same time depends on initial values of these parameters. This indicates that this fitting process could lead to the different values of the parameters, which simultaneously satisfy the approximation conditions. The sentence: “This task produced several results and was not mathematically distinctive, as shown by several approximations that were carried out here” is reformulated in the manuscript: “This task is mathematically no single tasking and depends on the assumed initial values of constitutive parameters.”

5.       What is “glextr” it?!

      (2) is the logical equation of the formulated postulate, where “glextr” means “global extremum” of the function f (X).

Reviewer 2 Report

An interesting article about significant cognitive problems. The methodology for identifying the parameters of constitutive equations was presented clearly. The test set-up chapter was unnecessary expanded. What important information are given for the reader in case of Figs. 1 and 2? I suggest removing them to leave space for other data. The compression tests were created for a range of strain rates and temperatures. It is worth presenting these results in the form of appropriate charts. For the cutting tests an indexable inserts (SNMG-SM-1105), were used as inserts. Is this an insert with flat rake plane? Or with profiled? I think that with profiled and when yes, why you used a flat model for FEM?

Clearly missing the graphs for different strain rates for the highest values. In Fig.3 you presenting the results for titanium alloy but for low strain rates. This parameters are very important for JC model definition and for good compatibility with results obtained by FEM.

On Fogs 5-8 you presented the cutting test results. What accuracy did you achieve?

In Table 2 you wrote the constitutive parameters established with the second algorithm. Do you also have the value of these parameters for the first algorithm? It would be interesting to analyze the differences in these parameters and their impact on the results of FE simulations.

In Table 4 are parameters for the cutting tests or for simulations? Table 5 presenting the comparison of predicted and experimental forces in orthogonal cutting. Why did not you perform a graphical analysis of these results (charts)? On the basis of such results, you can better write conclusions and more accurately evaluate the results of the simulation. Information about the fact that a good agreement was obtained without reference to other simulation results has a low cognitive value. You proposed two algorithms for determining JC parameters. Why not compare their effects in simulation? I believe that such a solution would allow to better assess the correctness of the proposed algorithms.

Author Response

Answers to the comments of Reviewer #2:

The authors are very grateful to Reviewer # 2 for meticulously reviewing and interpreting the content of the paper.

Reviewer #2: An interesting article about significant cognitive problems. The methodology for identifying the parameters of constitutive equations was presented clearly.

1.       “The test set-up chapter was unnecessary expanded. What important information are given for the reader in case of Figs. 1 and 2? I suggest removing them to leave space for other data.”

The Fig. 1 and Fig. 2 are removed.

2.       “The compression tests were created for a range of strain rates and temperatures. It is worth presenting these results in the form of appropriate charts.”

The results of the compression test are shown in Figure 2 as the dependence of strain and strain on stress. The effect of the strain rate in the investigated range of 0.05 s-1 to 10 s-1 is barely noticeable (see page 6, lines 224 to 228). The effect becomes significantly noticeable with greater strain rate. However, this is limited due to the range of the used test stand (the maximum strain rate of the test stand is 10 s-1).

3.       “For the cutting tests an indexable inserts (SNMG-SM-1105), were used as inserts. Is this an insert with flat rake plane? Or with profiled? I think that with profiled and when yes, why you used a flat model for FEM?”

The insert SNMG-SM-1105 have a flat rake face. The inserts are chosen with a flat rake face to eliminate the additional effects of the special profiles during machining.

4.       “Clearly missing the graphs for different strain rates for the highest values. In Fig.3 you presenting the results for titanium alloy but for low strain rates. This parameters are very important for JC model definition and for good compatibility with results obtained by FEM.”

The study of material behavior with high values of strain rates, which reach the magnitudes of 106 s-1 and more during cutting, should be carried out on appropriate and very expensive equipment (eg. split-Hopkinson pressure bar). We have used another way to achieve such high strain rates - even the cutting process. We think that the real process of strain, strain rate and temperature as well as their interaction should be defined from the cutting process for the determination of constitutive parameters to modeling of cutting. Especially this interaction is important for the correct modeling of the cutting process.

5.       “On Fogs 5-8 you presented the cutting test results. What accuracy did you achieve?”

For cutting force and chip compression ratio of AISI 1045 was the average measurement uncertainty 8% and 6% corresponding, for cutting force and chip compression ratio of titanium alloy Ti-1023 - 11% and 10% corresponding. The information is added to manuscript.

6.       “In Table 2 you wrote the constitutive parameters established with the second algorithm. Do you also have the value of these parameters for the first algorithm? It would be interesting to analyze the differences in these parameters and their impact on the results of FE simulations.”

The constitutive parameters determined by the first algorithm are inserted in the Table 2. The effect of the two algorithms is analyzed. The results are shown in Fig. 10 and in the text of the manuscript.

7.       “In Table 4 are parameters for the cutting tests or for simulations?”

Table 4 presents the cutting parameters used for verifying the FEM cutting model. Both cutting tests and simulations were performed on these parameters.

8.       “Table 5 presenting the comparison of predicted and experimental forces in orthogonal cutting. Why did not you perform a graphical analysis of these results (charts)? On the basis of such results, you can better write conclusions and more accurately evaluate the results of the simulation. Information about the fact that a good agreement was obtained without reference to other simulation results has a low cognitive value.”

For the representation of the data from Table 5, 8 different charts are required: shear angle, strain, cutting temperature and cutting force respectively for the steel AISI 1045 and titanium alloy T-1023. In addition, the corresponding deviation is given in Table 5, which presents the quality of the FE cutting model. We think the table presents the large amount of data recovered in a compact form better than the multiple charts. However, if the reviewer insists on that these results must be shown with charts, the authors will also add corresponding 8 charts.

9.       “You proposed two algorithms for determining JC parameters. Why not compare their effects in simulation? I believe that such a solution would allow to better assess the correctness of the proposed algorithms.”

This is done. The manuscript is extended with the results.

Round  2

Reviewer 1 Report

The manuscript was improved according to the sent comments.

Reviewer 2 Report

After the revision, the article looks much better. For example, Fig.10 facilitates the analysis of results. I still suggest to work in the future on good tests for higher deformation speeds.